# Hepatic PEMT Expression Decreases with Increasing NAFLD Severity

**DOI:** 10.3390/ijms23169296

**Published:** 2022-08-18

**Authors:** Ignazio S. Piras, Anish Raju, Janith Don, Nicholas J. Schork, Glenn S. Gerhard, Johanna K. DiStefano

**Affiliations:** 1Translational Genomics Research Institute, Phoenix, AZ 85004, USA; 2Department of Medical Genetics and Molecular Biochemistry, Lewis Katz School of Medicine at Temple University, Philadelphia, PA 19122, USA

**Keywords:** hepatic steatosis, gene expression, choline, nonalcoholic steatohepatitis, menopause, obesity, genetic variants

## Abstract

Choline deficiency causes hepatic fat accumulation, and is associated with a higher risk of nonalcoholic fatty liver disease (NAFLD) and more advanced NAFLD-related hepatic fibrosis. Reduced expression of hepatic phosphatidylethanolamine N-methyltransferase (*PEMT*), which catalyzes the production of phosphatidylcholine, causes steatosis, inflammation, and fibrosis in mice. In humans, common *PEMT* variants impair phosphatidylcholine synthesis, and are associated with NAFLD risk. We investigated hepatic *PEMT* expression in a large cohort of patients representing the spectrum of NAFLD, and examined the relationship between *PEMT* genetic variants and gene expression. Hepatic *PEMT* expression was reduced in NAFLD patients with inflammation and fibrosis (i.e., nonalcoholic steatohepatitis or NASH) compared to participants with normal liver histology (β = −1.497; *p* = 0.005). *PEMT* levels also declined with increasing severity of fibrosis with cirrhosis < incomplete cirrhosis < bridging fibrosis (β = −1.185; *p* = 0.011). Hepatic *PEMT* expression was reduced in postmenopausal women with NASH compared to those with normal liver histology (β = −3.698; *p* = 0.030). We detected a suggestive association between rs7946 and hepatic fibrosis (*p* = 0.083). Although none of the tested variants were associated with hepatic *PEMT* expression, computational fine mapping analysis indicated that rs4646385 may impact *PEMT* levels in the liver. Hepatic *PEMT* expression decreases with increasing severity of NAFLD in obese individuals and postmenopausal women, and may contribute to disease pathogenesis in a subset of NASH patients.

## 1. Introduction

Nonalcoholic fatty liver disease (NAFLD) encompasses a wide spectrum of histological conditions arising from the excessive accumulation of fat in liver. NAFLD is the most common chronic liver disease [1], and is often progressive. Many NAFLD patients also develop steatohepatitis and fibrosis, collectively representing nonalcoholic steatohepatitis (NASH), an advanced form of NAFLD that is associated with increased liver-related morbidity and mortality [2]. NASH patients have a greater risk of developing cirrhosis and hepatocellular carcinoma [3].

Choline is required for human health [4]. In the liver, choline plays important roles in lipoprotein transport, methyl group metabolism, and cell membrane signaling, and is required for the synthesis of membrane phospholipids, including phosphatidylcholine, lysophosphatidylcholine, choline plasmalogen, and sphingomyelin [4]. The Adequate Intake (AI) for choline is 550 mg/day for men and 425 mg/day for women, although a significant number of individuals do not consistently meet these recommended levels [5,6,7]. Low dietary choline intake has been associated with higher risk of NAFLD and worse NASH-related fibrosis [5,8]. Studies of dietary depletion of choline in humans have demonstrated significant liver dysfunction in otherwise healthy individuals in as little as two weeks [9,10,11], and a choline-deficient diet is commonly used in mouse models of human NAFLD [12].

Low levels of choline are produced endogenously through the methylation of phosphatidylethanolamine to phosphatidylcholine by the action of phosphatidylethanolamine N-methyltransferase (*PEMT*) [13]. *PEMT^−/−^* knockout mice fed a diet high in fat and sucrose rapidly develop hepatic steatosis, inflammation, and fibrosis, which is reversed by dietary choline supplementation [14,15,16,17,18]. *PEMT^−/−^* mice are also protected from high-fat-diet-induced obesity and insulin resistance [18,19]. In humans, common *PEMT* variants impair phosphatidylcholine synthesis, and are associated with NAFLD risk [20,21], which is exacerbated by low dietary choline intake [9,22]. Hepatic *PEMT* expression was observed to be lower in NASH patients compared to those with simple steatosis, and was significantly correlated with platelet counts, which typically decline with advancing fibrosis [14]. However, investigation into hepatic *PEMT* expression and NAFLD has been limited by sample size (total N = 34) and a focus on NAFLD extremes without a control group of individuals with normal liver histology [14]. To our knowledge, the relationship between *PEMT* genotypes and hepatic *PEMT* expression remains unexplored. The purpose of this study was to extend previous findings by assessing *PEMT* gene expression in a large cohort of patients representing the spectrum of NAFLD, and examine the relationship between *PEMT* genetic variants and hepatic gene expression.

## 2. Results

### 2.1. Patient Characteristics

The majority (80.1%) of study participants were females of European ancestry with a mean (± SD) age of 45.4 ± 11.3 and body mass index (BMI) of 47.4 ± 9.6 (Table 1). Individuals were grouped by hepatic histological classification (See Methods). For individuals with steatosis, 34 were grade 2 and 16 were grade 3, while 75% of participants had grade 1 lobular inflammation and the remainder were grade 2. Individuals with fibrosis tended to be older, had a higher BMI, and were more likely to have type 2 diabetes (T2D). The distribution of fibrosis was 38% bridging fibrosis, 28% incomplete cirrhosis, and 34% grade 4 cirrhosis.

### 2.2. Hepatic PEMT Expression Is Significantly Reduced in Individuals with Fibrosis

A previous study demonstrated that hepatic *PEMT* expression was significantly lower in Japanese NASH patients (n = 25) compared to individuals with simple steatosis (n = 9) [14]. To extend these findings to a larger, more comprehensively characterized cohort of Caucasian individuals (Table 1), we first compared *PEMT* expression in the group of patients with inflammation and fibrosis (n = 105), which we define as NASH, to those with normal liver histology (n = 36). We observed significantly lower *PEMT* expression with NASH (β = −1.497; *p* = 0.005; Figure 1A), consistent with the earlier report [14]. In pairwise comparisons of *PEMT* expression levels in participants with normal liver histology, we found significantly reduced levels of hepatic *PEMT* expression in NAFLD patients with inflammation (β = −1.476; *p* = 0.015) and fibrosis (β = −1.504; *p* = 0.013), but not with steatosis, although a similar trend was observed (β = −0.550; *p* = 0.222) (Figure 1B). Additionally, we observed a significant decrease in *PEMT* expression across the histological spectrum, from normal liver histology to fibrosis (β = −0.909; *p* =2.2 × 10^−4^). To determine whether *PEMT* expression decreased with increasing severity of hepatic fibrosis, we examined RNA-sequencing-derived expression data among different fibrosis stages, and observed a significant trend in decreasing *PEMT* levels with cirrhosis < incomplete cirrhosis < bridging fibrosis (β = −1.185; *p* = 0.011). The correlation remained significant with the inclusion of expression data from control samples (β = −0.152; *p* = 0.001; Figure 1C).

*PEMT* gene expression is regulated by estrogen [23,24], and the effects of a choline-deficient diet may be exacerbated in women who experience diminished estrogen production as a result of menopause [8,10,22]. To determine whether postmenopausal women with NASH have altered *PEMT* levels, we compared postmenopausal women with NASH (n = 31) to those with normal liver histology (n = 8). We observed significantly decreased hepatic *PEMT* expression in women with NASH (β = −3.698; *p* = 0.030), consistent with the trend observed in the overall cohort (Figure 2A). Although limited by the number of individuals in each group, we also detected a significant decrease in *PEMT* expression (β = −3.599; *p* = 0.008) with increasing stage of fibrosis in postmenopausal women (Figure 2B).

### 2.3. Fine Mapping Prioritizes rs4646385 for Functional Validation as the Lead SNP

To determine whether *PEMT* variants exert effects on gene expression, we accessed the GTEx V8 database of SNP genotypes and PEMT RNA expression in the 208 available healthy liver samples. We identified 20 SNPs that were significantly associated with *PEMT* expression (Appendix A), i.e., that were expression quantitative trait loci (eQTL). However, these SNPs showed varying degrees of linkage disequilibrium (LD) with one another, including some pairs exhibiting strong LD (Figure 3A). We therefore performed a fine mapping analysis of these variants, and found the highest posterior probability (Prob = 0.287) for rs4646385, suggesting that this SNP is the most likely causal variant (Figure 3B; Appendix A). No other eQTL showed a significantly elevated probability.

### 2.4. Common PEMT Variants Are Not Significantly Associated with NAFLD or Hepatic PEMT Expression

Because common single-nucleotide polymorphisms (SNPs) within the *PEMT* locus have been associated with risk of developing liver dysfunction in response to a choline-deficient diet [9,22] and susceptibility to NAFLD [20], including NAFLD in lean individuals [21], we investigated the association between three of the previously studied common SNPs (rs3760188-C/T, rs4646365-T/C, and rs7946-T/C) and hepatic fibrosis in our cohort. We observed a suggestive association between rs7946 and hepatic fibrosis, with risk allele (T) frequency calculated as 69.1% in the controls and 83.3% in the fibrosis group (*p* = 0.083). However, neither of the other variants were associated with fibrosis (Appendix A). No statistically significant evidence for an association between *PEMT* SNPs and fibrosis was observed in the subset of postmenopausal women (Appendix A). We also tested the association of these three variants with hepatic *PEMT* expression, and detected a suggestive association of rs3760188 with the effect allele (T) associated with higher *PEMT* expression (beta = 0.200, *p* = 0.092; Appendix A). No other associations were detected between *PEMT* alleles and *PEMT* expression, either in the total sample or the postmenopausal women group (Appendix A).

To determine whether a significant relationship between *PEMT* genotypes and NAFLD risk might be masked by the presence of obesity in our cohort, we also tested the association between *PEMT* variants and NAFLD in normal weight Caucasian individuals from the UKB. However, in our analyses, none of the variants assessed showed significant differences in allele frequency between cases and controls (Appendix A).

## 3. Discussion

Here we demonstrate that hepatic *PEMT* expression decreases with increasing severity of NASH in a large sample of individuals spanning the NAFLD spectrum. These findings are consistent with those reported by Nakatsuka et al. [14], who observed significantly lower hepatic *PEMT* expression in NASH patients compared to those with simple steatosis, suggesting that diminished *PEMT* expression may contribute to worse liver manifestations in a subset of NAFLD patients. Accordingly, we found significantly lower hepatic *PEMT* expression in NASH patients with cirrhosis versus those with milder liver fibrosis. In Japanese NASH patients, *PEMT* mRNA expression was significantly correlated with platelet counts, which decline with the progression of fibrosis, an observation that supports our findings. Our results obtained from a cohort of individuals with severe obesity contrast with earlier work reporting a significant correlation between low *PEMT* expression and lower BMI [14]. Furthermore, *PEMT*^−/−^ mice are resistant to high-fat-diet-induced obesity [18,19]. Our observations, therefore, indicate that even in the presence of obesity, decreased *PEMT* levels may contribute to NAFLD severity.

The study sample comprised a relatively large number of well-characterized individuals with biopsy-proven NAFLD, as well as a control group of subjects with normal liver histology. Gene expression data and *PEMT* genotypes were available for most of the individuals in the cohort. We were also able to replicate our findings in independent cohorts, which lends rigor to the results. Despite the significance of these findings, our results do not allow us to determine whether reduced *PEMT* expression is a cause or a consequence of NAFLD. However, because the PEMT pathway is the sole source of endogenous hepatic choline production, we speculate that reduced hepatic *PEMT* expression may contribute to the development and progression of NAFLD and NASH in susceptible individuals through insufficient generation of endogenous choline, which may have greater significance for those not meeting daily dietary choline needs. Triacylglycerol is synthesized in the liver, and is carried to other tissues by very low-density lipoprotein (VLDL). VLDL formation is dependent on the synthesis of new phosphatidylcholine molecules, and in the absence of phosphatidylcholine, VLDL cannot be generated and fat droplets accumulate in hepatocytes [13,25,26]. In addition to its contribution to VLDL synthesis, choline is involved in host-gut microbiota interactions and one-carbon metabolism, both of which have been linked to NAFLD pathogenesis. It is clear that choline is required for normal liver function, as choline-deficient diets [4,27] or diminished PEMT activity [15,16,28] results in hepatic steatosis. Moreover, choline supplementation prevents the development of steatosis and fibrosis caused by high sucrose diet in rats [29] and alleviates hepatic steatosis in individuals receiving long-term parenteral nutrition [30], suggesting that, at least for some individuals, choline replenishment may be an appropriate treatment to prevent the progression of NAFLD.

*PEMT* gene expression is regulated by estrogen [23,24]. Women of reproductive age appear to be more resistant to liver damage incurred by dietary choline restriction compared to men and postmenopausal women [8,10,12]. For example, Fischer et al. [10] reported that when deprived of dietary choline (i.e., <50 mg choline/day) for up to 42 days, 80% of postmenopausal women developed liver dysfunction, compared to only 38% of premenopausal women and 40% of men. Postmenopausal women supplemented with exogenous estrogen were also found to be less likely to develop liver dysfunction in response to a low choline diet [9]. We therefore considered the possibility that estrogen deficiency would correspond with reduced hepatic *PEMT* expression. In our cohort, postmenopausal women with NASH had significantly reduced hepatic *PEMT* expression compared to those with normal liver histology, suggesting that decreased *PEMT* expression may be a feature of liver dysfunction in women with estrogen deficiency. As in the overall cohort, we detected lower *PEMT* expression in postmenopausal women with fibrosis compared to milder NASH stages, although these analyses were limited by the number of individuals in the different NAFLD stages. Together with earlier studies, the current findings indicate that postmenopausal women with low estrogen levels may be at higher risk for NAFLD due to reduced *PEMT* expression, which would be exacerbated by low dietary choline intake [8,9,10,22].

Previous studies have reported effects of several *PEMT* variants. The variant genotype of rs7946, which encodes a Val-to-Met substitution at residue 175 of the human PEMT protein, is significantly more frequent in NAFLD patients compared to non-NAFLD controls [20,31,32]. Interestingly, expression of the two forms of PEMT in a hepatoma cell line demonstrated a 40% lower specific activity of the Met isoform relative to the Val isoform [20], indicating a functional consequence of the substitution. Other *PEMT* variants rs1235817, rs4646343, and rs3760188 were strongly associated with developing organ dysfunction in women who underwent choline depletion, while postmenopausal women who were carriers for the effect allele of other variants, rs1531100 and rs4646365, were at higher risk of liver damage in the presence of a low choline diet [22].

We observed a suggestive association between rs7946 and hepatic fibrosis, with the risk allele frequency higher in NASH patients with fibrosis compared to individuals with normal liver histology. However, we did not detect a significant effect of any of the other variants in our cohort, nor did we detect significant association between *PEMT* variants and hepatic *PEMT* expression. It is possible that (1) the variants analyzed in this study do not play a role in the transcriptional regulation of *PEMT* expression, or (2) an association between *PEMT* variants and NAFLD was masked by an adequate dietary choline intake in our cohort, which was drawn from a bariatric surgery population with severe obesity in whom dietary information was not available. Guerrerio et al. [8] reported that dietary choline intake increased with BMI, and observed a significant correlation between choline intake and total daily calorie consumption. These factors may be relevant for our findings from a cohort of individuals with obesity. On the other hand, decreased *PEMT* activity caused liver dysfunction in *ob/ob mice*, and significant alterations to the hepatic phosphatidylcholine: phosphotidylethanolamine molar ratio during obesity or overnutrition appear to promote NASH progression [19].

To address the possibility that *PEMT* genotypes confer NAFLD risk only in lean individuals, we tested the association between risk alleles and NAFLD in normal weight individuals from the UKB. Although a previous study reported a threefold higher risk of NAFLD in lean Asian Indian carriers of the rs7946 risk genotype [21], we did not find evidence for statistically significant association between any of the *PEMT* variants and NAFLD in 21 Caucasians with a lean BMI (i.e., BMI ≤ 25 kg/m^2^). Zeisel [32] noted that the presence of the variant rs7946 allele may not be sufficient to cause fatty liver, as many individuals with the risk allele have normal levels of liver fat, and instead, posited that the presence of the variant allele may slow triacylglycerol export from the liver. Therefore, fatty liver would develop only when triacylglycerol production is high, such as under conditions of excess energy intake [32], which would overwhelm hepatic fat export in individuals with the risk allele. If true, then individuals with fatty liver may be more likely to carry the variant allele, while those with the variant allele may not necessarily develop fatty liver. While our findings are consistent with this hypothesis, additional genetic studies are needed to provide confirmation.

We explored the association between 20 significant *PEMT* SNPs, curated from the GTEx database, significantly associated with hepatic *PEMT* RNA expression as eQTLs at this locus. Of the variants found to be significantly associated with *PEMT* expression, surprisingly, none corresponded to those previously reported in the literature [20,21,22,31,32], consistent with our results for a subset of those previously reported SNPs. Because we also observed strong LD among eQTL SNPs, suggesting that there might be only one or a few causal SNPs with the others in LD, rs4646385 may be the key *PEMT* variant. Efforts to prioritize this variant for functional validation are thus warranted.

NASH develops as a result of multiple, independent pathways with diverse lifestyle, environmental, and genetic components; dietary choline deficiency represents one such pathway. Interestingly, dietary choline requirements appear to be variable among individuals. One study observed that some subjects required more than the recommended AI for choline to avoid the development of metabolic dysfunction, while others had a much lower requirement [10]. Estrogen status was found to account for some of this variability [10]. Furthermore, while plasma concentrations of choline decreased in response to a low choline diet, the reduction was not highly correlated with susceptibility to organ dysfunction, suggesting that plasma choline levels, per se, are not useful for predicting choline-deficiency-related liver dysfunction.

Chronic choline deficiency may be relevant for a specific subset of NAFLD/NASH patients, specifically individuals with habitually low dietary choline intake (e.g., vegetarians and vegans), women with estrogen deficiency, or carriers of *PEMT* risk variants (Figure 4). Endogenous choline production may partially compensate for reduced dietary choline intake [33]. The absence of data on dietary choline intake in the current study prevents us from exploring this relationship and, therefore, limits the interpretation of our findings. Another limitation of this study is the age-delimited definition of menopausal status. The regulation of *PEMT* by estrogen implies that physiological estrogen levels may be important to measure in studies of PEMT function.

In summary, we observed lower hepatic *PEMT* expression with increasing severity of liver damage in NAFLD patients with obesity. Menopausal status and genotype may also affect *PEMT* expression. These results suggest that a subset of NAFLD/NASH patients may benefit from dietary choline supplementation.

## 4. Materials and Methods

### 4.1. Human Subjects

Participants selected for this study were enrolled in the Bariatric Surgery Program at the Geisinger Clinic Center for Nutrition and Weight Management. Individuals were between the ages of 21 and 70 years. Clinical data collected on patients were obtained as part of standard-of-care and stored in a research database. At the time of surgery, liver wedge biopsies were routinely performed as standard-of-care to assess liver histology. Pathologists read all liver biopsy slides using established NASH CRN criteria, as previously described [34]. Masson’s trichrome was employed to detect perisinusoidal fibrosis. Histological evaluations included normal liver histology (absence of inflammation, fibrosis, and ≤5% steatosis), steatosis (grade 1: 5–33% parenchymal involvement by steatosis, grade 2: >33–66% parenchymal involvement by steatosis, and grade 3: >66% parenchymal involvement by steatosis), inflammation (grade 1: <2 lobular inflammatory loci per 200× field, grade 2: 2–4 lobular inflammatory loci per 200× field), and fibrosis (grade 1: perisinusoidal, grade 2: perisinusoidal and periportal, grade 3: bridging fibrosis, grade 3/4: incomplete cirrhosis, and grade 4: cirrhosis). Patients with histologic or serologic evidence for other chronic liver diseases or significant alcohol use were excluded from this study. Clinical data, including demographics, clinical measures, ICD-9 codes, medical history, medication use, and common lab results were available for all study participants [35].

All study participants provided written informed consent for research, which was conducted according to The Code of Ethics of the World Medical Association (Declaration of Helsinki). The Institutional Review Boards of Geisinger Health System, Translational Genomics Research Institute, and Temple University School of Medicine approved the research protocol.

### 4.2. PEMT Hepatic Expression Analysis

RNA sequencing data were processed using the Illumina pipeline CASAVA v1.8.4 to generate FASTQ files; low-quality reads were removed during data processing. Alignment was conducted using Bowtie [36] against the human assembly GRCh37, and quantification of the number of reads for gene was conducted using the HTseq tool [37]. To make the data suitable for regression analysis, we normalized the dataset using the voom algorithm [38] and adjusted for batch effect using ComBat [39]. Outliers were identified by Principal Component Analysis (PCA). To investigate the relationship between *PEMT* RNA expression and histological grade, we modeled an ordinal logistic regression considering the grade as the dependent variable, gene expression as a predictor, and adjusted for sex, age, type 2 diabetes (T2D) status, and fasting triglyceride concentration. We conducted the same analysis in postmenopausal women, defining menopausal state by an age ≥ 51 years, which is the average age at menopause in the United States [40], as information pertaining to reproductive status was not available for this cohort. Associations with *p* < 0.05 were considered statistically significant.

### 4.3. PEMT Variant x PEMT Expression Analysis

To assess the relationship between *PEMT* variants and *PEMT* expression, we utilized data from the GTEx V8 database [41]. Liver-tissue-specific expression quantitative trait loci (eQTL) associated with differential *PEMT* expression were filtered to identify significant variants based on the corresponding Storey q-value. After excluding nonsignificant variants, the final list consisted of 20 single-nucleotide polymorphisms (SNPs) that were significantly associated with differential *PEMT* expression. Pairwise linkage disequilibrium (LD) data for these variants was extracted from the European 1000 Genomes project [42]. We performed a fine mapping analysis to identify a potential lead or causal variant from the curated list using the probabilistic framework PAINTOR, integrating association strength and SNP correlations [43,44].

### 4.4. Association Analysis of PEMT Variants and NAFLD

The processing of SNP microarray data was described previously [34]. After quality control filtering, the initial dataset consisted of 156 individual samples for 547,937 SNPs. Of the 53 individuals with fibrosis with RNA profiling data, 39 of them were also represented in the SNP microarray dataset [34]; therefore, we extracted genotype data for the following *PEMT* polymorphisms, rs3760188 (C/T), rs4646365 (T/C), and rs7946 (T/C), in participants with fibrosis (cases; n = 39) and normal liver histology (controls; n = 34). All SNPs were located within the *PEMT* locus: rs3760188 (C/T) was located in 17:17487440 (intron 1), rs4646365 (T/C) was located in 17:17467783 (intron 2), and rs7946 (T/C) was located in 17:17409560 (exon 6, missense mutation). No other SNPs in *PEMT* were available in our dataset.

Association analysis between fibrosis cases and controls was conducted using R, modeling a logistic regression adjusting for sex, age, fasting triglyceride concentration, and diabetes status. Expression quantitative trait loci (eQTL) analysis between the three SNPs and *PEMT* expression was conducted modeling a linear regression with gene expression as the dependent variable and the genotypes as predictors, adjusting for sex, age, triglycerides, and diabetes status. We utilized the same approach for testing association in postmenopausal women. Associations with *p* < 0.05 were considered statistically significant.

We also investigated the association of *PEMT* SNPs in a second cohort comprised of lean individuals (BMI ≤ 25 kg/m^2^) from the UK Biobank (access date: 29 March 2022: http://www.ukbiobank.ac.uk/). We defined NAFLD cases using ICD codes (ICD-10 = K758, ICD-9 = 5718 or 5719). We extracted ICD codes (up to 29 March 2022) from inpatient and death records of UKB. We also searched for these ICD codes in the following UKB data fields: 40001, 40002, 40006, 41201, 41202, 41204, 41270, 40013, 41203, 41205, 4127. Subjects with other chronic liver diseases or significant alcohol use were excluded from our analyses. Because *PEMT* variants may have greater relevance in individuals of Northern European ancestry [32], we limited our analysis to white British individuals using the UKB data field 21000 (ethnic background). We removed first-degree relatives and those showing sex discrepancies between indicated sex (UKB field-31) and genetic sex (UKB field-22001). We selected normal-weight individuals based on a BMI threshold ≤ 25 kg/m^2^ using the UKB data field 23104, although in some cases, there were age gaps between BMI measurement and NAFLD diagnosis. After aggregating these data with the covariates and genotyped data, we identified 21 NAFLD cases and 132,084 NAFLD-negative controls (85,955 females, 46,150 males). For the genotype data, we used UKB imputed variants version 3, which are aligned to the + strand of the reference and with GRCh37 coordinates. We tested each of the following variants, rs12325817, rs4646343, rs3760188, rs7946, rs1531100, rs4646365 (all of them having imputed information score > 0.8), with a logistic regression model using PLINK 2.0 (access date: 29 March 2022: www.cog-genomics.org/plink/2.0/), having cases/controls as the dependent variable, genotype as the independent variable, and adjusted for age, sex, diabetes status, and first 10 principal components. We did not include triglyceride levels as a covariate because we observed (1) large differences in measures between visits for some individuals and (2) significant time gaps between the date of triglyceride measurement and the date of NAFLD diagnosis.

## Figures and Tables

**Figure 1 ijms-23-09296-f001:**
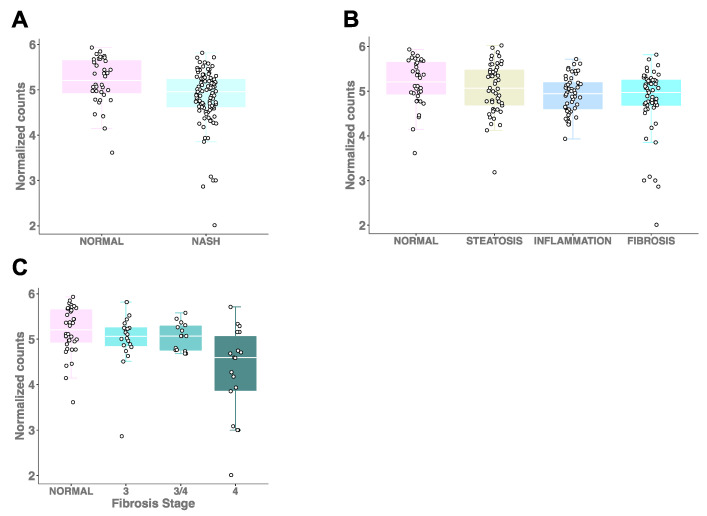
Hepatic *PEMT* expression decreases with increasing severity of NAFLD. (**A**) *PEMT* expression in individuals with NASH (inflammation and fibrosis) and those with normal liver histology (β = −1.497; *p* = 0.005). (**B**) *PEMT* expression levels in participants spanning the histological spectrum of NAFLD (β = −909; *p* = 2.2 × 10^−4^). The expression levels were significantly different when comparing inflammation and fibrosis groups with controls (β = −1.476; *p* = 0.015) and (β = −1.504; *p* = 0.013, respectively), but not when comparing steatosis with controls (β = −0.550; *p* = 0.222). (**C**) *PEMT* expression in different fibrosis stages and controls (β = −0.152; *p* = 0.001).

**Figure 2 ijms-23-09296-f002:**
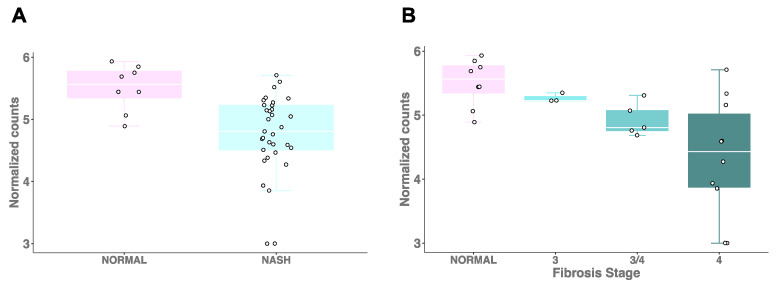
Hepatic *PEMT* expression is reduced in postmenopausal women with NAFLD and fibrosis. (**A**) *PEMT* expression in postmenopausal women with inflammation and fibrosis (i.e., NASH (n = 31) and normal liver histology (n = 8). (**B**) *PEMT* expression in postmenopausal NAFLD patients with varying degrees of fibrosis.

**Figure 3 ijms-23-09296-f003:**
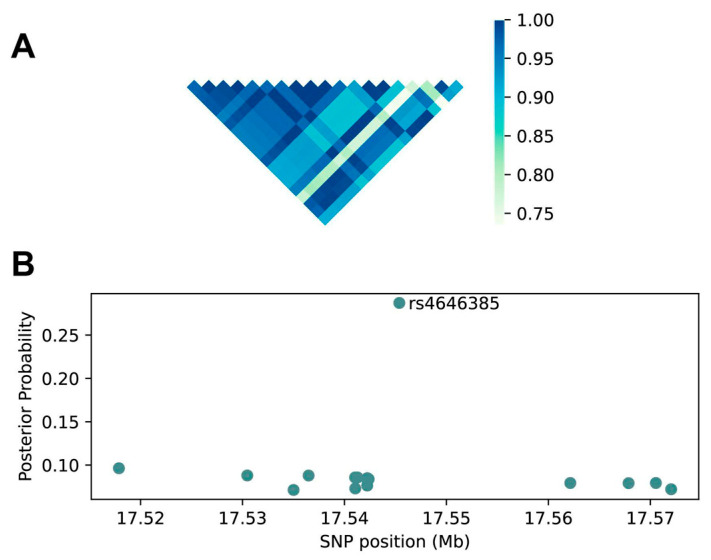
Fine mapping prioritizes rs4646385 as a potential lead SNP. (**A**) LD between significant *PEMT* eQTLs. Each square represents the LD value between two SNPs pairwise comparisons. (**B**) Estimated SNP posterior probabilities with SNP genome coordinates. Each point represents a different SNP curated from the GTEx database.

**Figure 4 ijms-23-09296-f004:**
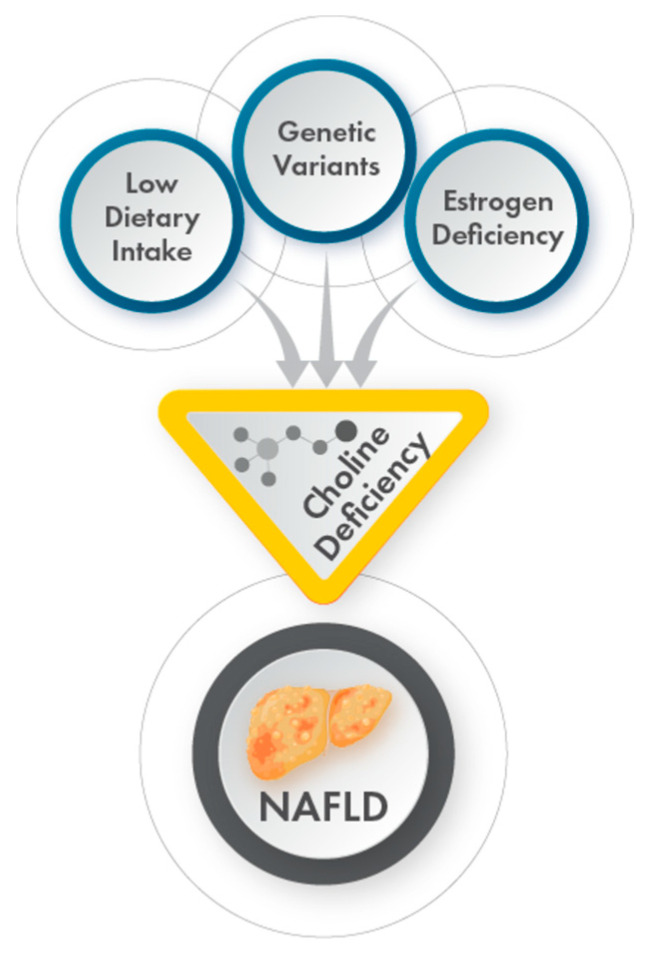
Diets low in choline, variants in genes involved in endogenous choline production, and estrogen deficiency may trigger a chronic choline-deficient state, which may increase the risk of developing NAFLD. Individuals possessing more than one of these factors are expected to be at greater risk of developing a choline-deficient state. Other factors, such as pregnancy and gut dysbiosis, may also affect choline bioavailability.

**Table 1 ijms-23-09296-t001:** Characteristics of study cohort.

	Normal Histology *	Steatosis **	Inflammation ***	Fibrosis ****
**N**	36	50	52	53
**M/F**	5/31	11/39	8/44	14/39
**Age (y) ± SD**	44.6 ± 9.8	44.0 ± 11.1	43.7 ±12.7	49.1 ± 10.3
**BMI (kg/m^2^) ± SD**	43.4 ± 6.2	46.7 ± 10.0	48.7 ± 7.8	49.5 ± 11.1
**T2D (N)**	9	18	14	36

* Absence of inflammation, fibrosis, and ≤5% steatosis; ** grade 2: >33–66% parenchymal involvement by steatosis, and grade 3: >66% parenchymal involvement by steatosis; *** grade 1: <2 lobular inflammatory loci per 200× field), grade 2: 2–4 lobular inflammatory loci per 200× field); **** grade 3: bridging fibrosis, grade 3/4: incomplete cirrhosis, and grade 4: cirrhosis.

## Data Availability

Not applicable.

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
