# Peer review of "Hepatic PEMT Expression Decreases with Increasing NAFLD Severity"

_ijms, 2022, doi:10.3390/ijms23169296_

Round 1

Reviewer 1 Report

This study examined the relationship between PEMT expression and NAFLD and NASH. I guess that SNPs
found in this study are novel
ty of this study. If it is right, this point should be emphasized. The authors claimed
that “PEMT genotypes and hepatic
PEMT expression remains unexplored” in introduction. However, the
authors also described as follows; “common PEMT variants impair phosphatidylcholine synthesis and are

associated with NAFLD risk”, that is inconsistent. Description of the aim of study shoul
d be match to results
and conclusion to make your novel finding
s clear. In addition, there are several concerns for publication as
follows.
Overall, information about experiments and samples is lacking.
1.
What are the differences of the result showing in TableS4 and TableS1 to S3. It is difficult to distinguish.
What I mean is the differences
of the experiment between section 2.3 and 2.4.
2.Is it impossible to
add the data which is derived from rs4646385 to TableS1 to S3? If possible, the data should
be added
since it is causal SNP.
3
.There are four groups including Normal, NASH, Steatosis and Inflammation in Fig1. How each group was
defined should be described
in method section. Especially, was “Normal” derived from healthy volunteer? The
origin should
be described clearly.
4
. How were normalized counts calculated in Fig1and 2? It should be described in method section.
5
. In Fig 2A, detailed information of each group should be described as like Table1.
6
. Parameters reflecting hepatic state including grade of NAFLD or NASH should be added to Table 1 as detailed
as possible.
It is essential. This is also applicable to No. 5 comment.
7
. In Fig.S3, where are these alleles located? In addition, why were these three alleles focused on? Is there no
possibility that
other alleles exert effect? There is no explanation.
8
. There are no description about statistical analysis. Section of statistical analysis and detailed explanation
about analysis
should be added.
9
. Marks showing statistical differences should be added to each figure consistently. In addition, statistical
differences
between all groups should be examined. If the authors think that there is no need to compare all
groups, rational explanation is necessary.

7. I’m not sure the necessity of Fig
S1. It can be unified into Fig1C.
8. Information of Fig3A is fatally lacking. What each dot indicate
s should be clear.
9
. Did you check submission rules? Tables, figures and legends are to be exerted to manuscript appropriately by
authors.

1
0. Resolution of Fig2 is low.

Author Response

Comments and Suggestions for Authors

This study examined the relationship between PEMT expression and NAFLD and NASH. I guess that SNPs found in this study are novelty of this study. If it is right, this point should be emphasized. The authors claimed that “PEMT genotypes and hepatic PEMT expression remains unexplored” in introduction. However, the authors also described as follows; “common PEMT variants impair phosphatidylcholine synthesis and are associated with NAFLD risk”, that is inconsistent. Description of the aim of study should be match to results and conclusion to make your novel findings clear. In addition, there are several concerns for publication as
follows. Overall, information about experiments and samples is lacking.

We appreciate the apparent inconsistencies noted by the reviewer. To clarify, the primary novelty of this study lies in the fine-grained analysis of PEMT RNA expression across a large sample size spanning the NAFLD histological spectrum, including normal histology, that we then mapped to SNPs to identify eQTL. Our observations indicate that even in the presence of obesity, decreased PEMT levels may contribute to NAFLD severity. To our knowledge, this relationship has not yet been reported in the literature.

While it is true that many studies in animal models and human subjects have shown a relationship between PEMT SNPs and either phosphatidylcholine synthesis and/or NAFLD risk, ours is the first to analyze the interaction between PEMT SNPs and hepatic PEMT RNA expression.

We apologize for lack of detail and provide more Information about the experiments and samples now detailed in the Material and Methods section.

  1. What are the differences of the result showing in TableS4 and TableS1 to S3. It is difficult to distinguish. What I mean is the differences of the experiment between section 2.3 and 2.4.

We apologize for the lack of clarity. Table S1 shows the posterior probability results for PEMT SNPs obtained from the GTEx dataset. Table S2 shows the allele frequencies and test statistics of the three PEMT SNPs evaluated in the bariatric surgery cohort with respect to fibrosis status. Table S3 shows the results of the association between PEMT SNPs and PEMT gene expression in the bariatric surgery cohort. Table S4 describes the test statistics for PEMT SNPs that were genotyped in the UKB cohort, focusing on normal weight individuals with NAFLD. Each supplementary table contains unique information that is not interchangeable with data from the other tables. We have revised the table titles to reflect their content more clearly.

Similarly, Section 2.3 used RNA expression data from the publicly available Genotype-Tissue Expression (GTEx) database to determine whether SNP genotypes correlated with hepatic PEMT expression in the 208 available non-diseased liver samples. This allowed us to identify 20 SNPs that were significantly associated with PEMT expression that we then narrowed to a single lead SNP in an intron of PEMT. In Section 2.4 we analyzed previously studied PEMT SNPs for their association with hepatic PEMT RNA expression in the bariatric surgery cohort. As described below (point 7), we selected these SNPs because they were located in the PEMT gene, had a prior association with NAFLD, and were available in our microarray dataset (DiStefano et al., PMID: 25246029).

  1. Is it impossible to add the data which is derived from rs4646385 to Table S1 to S3? If possible, the data should be added since it is the causal SNP.

The frequency data shown in Tables S1, S2, and S3 were calculated from a simple list of genotypes from the two different groups compared with the statistical results shown for each comparison. Genotype data for these SNPs, but not other PEMT SNPs, were available in the two cohorts. Unfortunately, frequency data for rs4646385 was not available for either the bariatric surgery cohort or the UKB participants.

  1. There are four groups including Normal, NASH, Steatosis and Inflammation in Fig1. How each group was defined should be described in method section. Especially, was “Normal” derived from healthy volunteer? The origin should be described clearly.

We apologize for the lack of clarity. Each group was defined by histological criteria. As suggested by the reviewer, we have detailed the histological criteria in the Materials and Methods section. All liver biopsy samples were obtained from patients undergoing bariatric surgery as part of their standard of care, not from healthy volunteers. 

  1. How were normalized counts calculated in Fig1and 2? It should be described in method section.

Normalized counts were obtained using the “voom” algorithm, as noted on lines 320-321.

  1. In Fig 2A, detailed information of each group should be described as like Table1.

We apologize for the confusion.  The groups in Fig 2A correspond to the groups described in Table 1.  We have modified the figure legend and results section to improve clarity.

  1. Parameters reflecting hepatic state including grade of NAFLD or NASH should be added to Table 1 as detailed as possible. It is essential. This is also applicable to No. 5 comment.

We have added detail on the grading of NAFLD and NASH to the Table 1 legend. 

  1. In Fig.S3, where are these alleles located? In addition, why were these three alleles focused on? Is there no possibility that other alleles exert effect? There is no explanation.

We apologize for the confusion. As described in the Fig title (which is now Fig S2; line 387), this figure depicts PEMT expression levels across the three rs3760188 genotypes. In the text, we have referenced this figure on line 145: “We also tested the association of these three variants with hepatic PEMT expression, and detected a suggestive association of rs3760188, with the effect allele (T) associated with higher PEMT expression (beta = 0.200, p = 0.092; Fig S2).”

Locations for the three SNPs tested in the bariatric surgery cohort have been added to the Methods section 4.4. “Association analysis of PEMT variants and NAFLD” (lines 347-351). We selected these SNPs because they were located in the PEMT gene, had a prior association with NAFLD, and were available in our microarray dataset (DiStefano et al., PMID: 25246029).

  1. There are no description about statistical analysis. Section of statistical analysis and detailed explanation about analysis should be added.

All the statistics and bioinformatics analysis details were reported in paragraphs 4.2, 4.3, and 4.4. (lines 292 - 384). For example, in Section 4.2, RNA sequencing data were processed using the Illumina pipeline CASAVA v1.8.4 to generate FASTQ files; low-quality reads were removed during data processing. Alignment was conducted using Bowtie (33) against the human assembly GRCh37, and quantification of the number of reads for gene was conducted using the HTseq tool (34). To make the data suitable for regression analysis, we normalized the dataset using the voom algorithm (35) and adjusted for batch effect using ComBat (36). Outliers were identified by Principal Component Analysis (PCA). To investigate the relationship between PEMT RNA expression and histological grade, we modeled an ordinal logistic regression considering the grade as the dependent variable, gene expression as a predictor, and adjusted for sex, age, type 2 diabetes (T2D) status, and fasting triglyceride concentration. We conducted the same analysis in postmenopausal women, defining menopausal state by an age ≥ 51 years, which is the average age at menopause in the United States (37), as information pertaining to reproductive status was not available for this cohort. Associations with p < 0.05 were considered statistically significant.

  1. Marks showing statistical differences should be added to each figure consistently. In addition, statistical differences between all groups should be examined. If the authors think that there is no need to compare all groups, rational explanation is necessary.

We appreciate the reviewer’s comment. However, our analysis focused on the comparison with normal controls and in the detection of a trend across the fibrosis stages, the most clinically important aspect of NAFLD/NASH, thus we did not conduct pairwise comparisons between all groups.

  1. I’m not sure the necessity of FigS1. It can be unified into Fig1C.

We appreciate the suggestion of the reviewer. Accordingly, we have replaced the original Fig 1C with a new figure that now contains the information originally shown in Fig S1. Consequently, the other supplementary figures have been renumbered to account for this change.

  1. Information of Fig3A is fatally lacking. What each dot indicates should be clear.

We apologize for the confusion. We assume that the reviewer was referring to Fig 3B, because that is the one with the dots. Each point represents one of the PEMT SNPs curated from the GTEx database. For Fig 3A, each square represents the LD value between two SNPs in pairwise comparisons. We have modified the figure legend to make this information more accessible.

  1. Did you check submission rules? Tables, figures and legends are to be exerted to manuscript appropriately by authors.

The tables, figures, and legends have been inserted into the current version of the manuscript.

  1. Resolution of Fig2 is low.

The resolution of Fig 2 has been increased and the revised figure has been added to the manuscript.

Reviewer 2 Report

Reviewer’s comments

This article primarily focused on an inverse correlation between hepatic PEMT expression and disease activity in patients with NAFLD. This article has its own originality and seems to be very interesting. I appreciate the efforts by the authors. However, the methodology is not fully described in this study. The experimental designs are partially inappropriate. The interpretations of results obtained from this study seem to be insufficient. It is regrettable to say that this article is not acceptable for publication. Please refer to the comments shown below.

Major

#1. The authors should describe inclusion and exclusion criteria The process for patients assignment should be depicted using a figure. The total number of assigned patients and the number of control cases should be noted. Were the assigned patients NAFLD or NASH? According to Fig.1, assigned patients were NASH. Was it correct?

#2. The authors should clearly described which criteria were selected for the evaluation of hepatic fibrosis and steatosis in the enrolled patients (The classification proposed by Brunt and colleagues?).  

The method for estimating the disease activity should be also mentioned (NAFLD activity score?). In addition, patient number in each subgroup (F1, F2, F3, F4 or grade 1m grade2 grade 3) should be addressed.

#3. Were control cases selected among the patients who underwent bariatric surgery? I think that most of such patients were obese (not normal). It is not appropriate to select normal liver tissues among the patients who underwent such the operation.

#4 The methods for statistical analyses were missing in Materials and Methods. The definition of the statistical significance in this study should be also described.

#5. This study revealed that the level of hepatic PEMT was unrelated to the severity of hepatic steatosis. The authors should describe or speculate the reasons.

#6. The authors did not confirm the age of menopause in the assigned patients at all. It is one of the limitations in this study. If this study focuses on the menopause, male patients should be excluded in the statistical process. Other limitations should be also listed up in this study.

#7. Fig.4 is missing in the text.

Minor

#1. Nonalcoholic steatohepatitis and NASH are in duplicated as keywords.

#2. P-values should be addressed in Fig.1A through Fig.1C and Fig.S1

#3. The columns of F1 and F2 are missing in Fig.1C. and Fig. 2B

#4. The patient number of each allele should be also addressed in Fig. S3.

#5. The external fund R01 DK091601 should be described in detail.

Author Response

Reviewer’s comments

This article primarily focused on an inverse correlation between hepatic PEMT expression and disease activity in patients with NAFLD. This article has its own originality and seems to be very interesting. I appreciate the efforts by the authors. However, the methodology is not fully described in this study. The experimental designs are partially inappropriate. The interpretations of results obtained from this study seem to be insufficient. It is regrettable to say that this article is not acceptable for publication. Please refer to the comments shown below.

Major

#1. The authors should describe inclusion and exclusion criteria The process for patients assignment should be depicted using a figure. The total number of assigned patients and the number of control cases should be noted. Were the assigned patients NAFLD or NASH? According to Fig.1, assigned patients were NASH. Was it correct?

We apologize for the lack of clarity. Inclusion and exclusion criteria have been added (lines 292-309). Patients were not actually “assigned” to a group, but rather, all biopsy samples were obtained from patients undergoing bariatric surgery as part of their standard of care, and were classified based on histological findings. Patients spanned the NAFLD spectrum and included those with simple steatosis, steatohepatitis, and fibrosis. The total number of patients, including those with normal liver histology, are shown in Table 1.

#2. The authors should clearly described which criteria were selected for the evaluation of hepatic fibrosis and steatosis in the enrolled patients (The classification proposed by Brunt and colleagues?).  The method for estimating the disease activity should be also mentioned (NAFLD activity score?). In addition, patient number in each subgroup (F1, F2, F3, F4 or grade 1m grade2 grade 3) should be addressed.

The biopsy samples were histologically evaluated using NASH CRN criteria as detailed in the Materials and Methods section (line 292). We have also added details of the histological criteria in response to the reviewer’s concern (lines 299-305). Unfortunately, we were not able to calculate NAFLD activity score in this cohort. We have added a column to Table 1 to represent the data stated in the results. 

#3. Were control cases selected among the patients who underwent bariatric surgery? I think that most of such patients were obese (not normal). It is not appropriate to select normal liver tissues among the patients who underwent such the operation.

All liver biopsy samples were obtained from patients undergoing bariatric surgery as part of their standard of care, as some clinical programs do (please see Barbois et al. Benefit-risk of intraoperative liver biopsy during bariatric surgery: review and perspectives. Surg Obes Relat Dis. 2017 Oct;13(10):1780-1786. PMID: 28935200.). The bariatric surgery population offers a unique opportunity to obtain biopsied liver tissue in the absence of a clinical indication. As such, individuals with obesity, while not “normal”, often present with normal liver histology. As the reviewer knows, obtaining liver tissue from normal, healthy individuals, outside of the standard of care for bariatric surgery is unethical.

#4 The methods for statistical analyses were missing in Materials and Methods. The definition of the statistical significance in this study should be also described.

 All the statistical and bioinformatics analysis details were reported in paragraphs 4.2, 4.3, and 4.4. (lines 316 - 384). Significant results were considered when p < 0.05. We added this information to the Methods section.

#5. This study revealed that the level of hepatic PEMT was unrelated to the severity of hepatic steatosis. The authors should describe or speculate the reasons.

We did not assess the relationship between hepatic PEMT expression and severity of hepatic steatosis in our study. However, we did observe a trend toward decreased hepatic PEMT expression in steatosis, but at levels that failed to meet statistical significance.  While we do not know the reason for this, it is possible that reduced PEMT expression only becomes a significant factor in the presence of hepatocellular injury.

#6. The authors did not confirm the age of menopause in the assigned patients at all. It is one of the limitations in this study. If this study focuses on the menopause, male patients should be excluded in the statistical process. Other limitations should be also listed up in this study.

The age of menopause in the assigned patients is found in the Materials and Methods section, lines 325-328. We investigated the relationship between PEMT expression and menopause as a component of the study, but it was not the main focus of the study; therefore, it is appropriate that males were included in the statistical analysis.

Limitations of the study were described in the Discussion section, lines 273-277.

#7. Fig.4 is missing in the text.

Fig 4 was referenced in the Discussion section, line 271-272.

Minor

#1. Nonalcoholic steatohepatitis and NASH are in duplicated as keywords.

We have removed one of the terms.

#2. P-values should be addressed in Fig.1A through Fig.1C and Fig.S1.

We added the p-values and regression coefficients to the Fig 1 legend. Fig S1 was incorporated into Fig 1C, as suggested by another reviewer.

#3. The columns of F1 and F2 are missing in Fig.1C. and Fig. 2B

We did not include patients with F1 and F2 in our analysis.

#4. The patient number of each allele should be also addressed in Fig. S3.

In Fig S3, each circle corresponds to an individual with the genotype indicated on the x-axis. Thus, there are 14 TT, 34 TC, and 27 CC carriers shown in the figure.

#5. The external fund R01 DK091601 should be described in detail.

Typically, the grant number is appropriate for referencing in a manuscript. If the journal editor requests more detailed information, we will provide it.

Reviewer 3 Report

In this well, design, executed and presented work, authors investigated and report significant association between reduced expression of hepatic phosphatidylethanolamine N-Methyltransferase (PEMT) and severity of non-alcoholic fatty liver disease (NAFLD) especially in postmenopausal and obese patients. I have some comments below.

1.      Method/Result: I strongly recommend that the method section be moved so that it comes before the result. Because the manuscript is heavy with abbreviations most of which are spelled out in the method, these should be read earlier to improve readability and avoid confusion.

2.      Lines 92-96:  The statement is not clear and failed to justify the question being investigated (i.e., to assess the effect of oestrogen deficiency on PEMT expression and liver dysfunction) if both groups compared are postmenopausal, then is NASH a cause or effect of the observed significant PEMT under-expression?

Please clarify the statement.

Author Response

In this well, design, executed and presented work, authors investigated and report significant association between reduced expression of hepatic phosphatidylethanolamine N-Methyltransferase (PEMT) and severity of non-alcoholic fatty liver disease (NAFLD) especially in postmenopausal and obese patients. I have some comments below.

We appreciate the reviewer’s positive comments about our work.

  1. Method/Result: I strongly recommend that the method section be moved so that it comes before the result. Because the manuscript is heavy with abbreviations most of which are spelled out in the method, these should be read earlier to improve readability and avoid confusion.

According to the manuscript preparation guidelines of the journal, the Materials and Methods section is to follow the Discussion section. However, we have edited the manuscript to spell out the abbreviations as they first appear in the text to improve readability and avoid confusion, as suggested by the reviewer.

  1. Lines 92-96:  The statement is not clear and failed to justify the question being investigated (i.e., to assess the effect of oestrogen deficiency on PEMT expression and liver dysfunction) if both groups compared are postmenopausal, then is NASH a cause or effect of the observed significant PEMT under-expression?

Please clarify the statement.

We apologize for the lack of clarity in this section. We have clarified the statement according to the reviewer’s suggestion. With respect to the excellent question posed by the reviewer, our results do not allow us to determine whether NASH is a cause or an effect of the observed significant PEMT under-expression. However, previous work by Zeisel’s group would suggest that NASH is an effect of PEMT downregulation; however, additional experiments would be required for confirmation.

Round 2

Reviewer 1 Report

The draft has been well revised.